# Yttrium and Niobium Elements Co-Doping and the Formation of Double Perovskite Structure Ba_2_YNbO_6_ in BCZT

**DOI:** 10.3390/ma16114044

**Published:** 2023-05-29

**Authors:** Runyu Mao, Deyi Zheng, Qiyun Wu, Yuying Wang, Chang Liu

**Affiliations:** College of Materials and Metallurgy, Guizhou University, Guiyang 550025, China; m15564182208_1@163.com (R.M.); wqy18522022314@163.com (Q.W.); yuyingwang1026@163.com (Y.W.); lc971020@163.com (C.L.)

**Keywords:** BCZT piezoelectric ceramics, double perovskite structure, phase structure, piezoelectric properties

## Abstract

The (Ba_0.85_Ca_0.15_) (Ti_0.90_Zr_0.10_)O_3_ + *x* Y^3+^ + *x* Nb^5+^ (abbreviated as BCZT-*x*(Nb + Y), *x* = 0 mol%, 0.05 mol%, 0.1 mol%, 0.2 mol%, 0.3 mol%) lead-free piezoceramics samples were prepared by a traditional solid-state sintering method. And the effects of Yttrium and Niobium elements (Y^3+^ and Nb^5+^) co-doping on the defect, phase and structure, microstructure, and comprehensive electrical properties have been investigated. Research results show that the Y and Nb elements co-doping can dramatically enhance piezoelectric properties. It is worth noting that XPS defect chemistry analysis, XRD phase analysis and TEM results together show that a new phase of double perovskite structure Barium Yttrium Niobium Oxide (Ba_2_YNbO_6_) is formed in the ceramic, and the XRD Rietveld refinement and TEM results show the coexistence of the R-O-T phase. Both these two reasons together lead to significant performance improvements of piezoelectric constant (*d*_33_) and planar electro-mechanical coupling coefficient (*k*_p_). The functional relation between temperature and dielectric constant testing results present that the Curie temperature increases slightly, which shows the same law as the change of piezoelectric properties. The ceramic sample reaches an optimal performance at *x* = 0.1% of BCZT-*x*(Nb + Y), where *d*_33_ = 667 pC/N, *k*_p_ = 0.58, *ε*_r_ = 5656, tan*δ* = 0.022, *P*_r_ = 12.8 μC/cm^2^, *E*_C_ = 2.17 kV/cm, *T*_C_ =92 °C, respectively. Therefore, they can be used as potential alternative materials to lead based piezoelectric ceramics.

## 1. Introduction

Over the past years, lead-free piezoelectric ceramics is the focus of research in the field of functional materials. Researchers have carried out in-depth research on K_0.5_Na_0.5_NbO_3_ (KNN), Bi_0.5_Na_0.5_TiO_3_ (BNT), and BaTiO_3_ (BT) based lead-free piezoelectric ceramics, which have been furtherly applied to the field of electronics industry [1]. Among these kinds of piezoelectric materials, (Ba_0.85_Ca_0.15_) (Ti_0.90_Zr_0.10_)O_3_ (BCZT) ceramic is a typical BT-based lead-free piezoelectric ceramic, which is regarded as one of the candidate materials for replacing PZT due to its low porosity, good piezoelectric coefficient and low dielectric loss [2,3]. However, the piezoelectric coefficient, the Curie temperature of BCZT ceramic still needs further improvement [2].

To this end, lots of researches have been conducted to improve the piezoelectric performance of BCZT ceramics, including compound introduction and element doping [2,3]. Shu et al. found that introducing a new perovskite phase (BNT) into BCZT contributed to lattice distortion and optimized the piezoelectric properties of ceramics (such as *d*_33_ increases from 340 pC/N to 450 pC/N), but the Curie temperature drops from 82 °C to about 70 °C [4]. Doping elements may also improve the piezoelectric properties of piezoelectric ceramics, but decrease its Curie temperature [5,6,7]. For instance, BCZT-*x*Tm^3+^ ceramics obtained superior piezoelectricity at *x* = 0.5% (*d*_33_ = 532 pC/N, *d*_33_* = 645 p.m./V) [7]. But its Curie temperature is only 74 °C, which is lower than that of pure BCZT ceramics (*T*c = 85 °C).

However, some studies have shown that doping rare earth oxide Y_2_O_3_ can dramatically improve the electrical performance of BCZT-based ceramics without reducing the Curie temperature [8,9]. Mittal et al. [8] recorded that when 0.02 mol% Y_2_O_3_ was introduced into BCZT ceramics, the grain grew effectively, the microstructure of ceramics was improved, and the density was increased. The piezoelectric coefficient and electro-mechanical coupling coefficient were increased by approximately 70% and 18% respectively, and the Curie temperature was stabilized at around 90 °C. Li et al. reported that Y^3+^ doping can promote the formation of oxygen vacancy, resulting in the *T*_C_ of BCZT ceramics samples increases by 5–15 °C compared to the pure BCZT [9]. Pie et al. found that Nb^5+^ doping into BCZT-based ceramics helped to enhance the electrical performance [10,11]. This superior piezoelectric properties after Nb^5+^ doping resulted from the orthorhombic (*O*)—tetragonal (*T*) phase coexistence and the tendency of ferroelectric domains to maintain a miniaturized domain type, which facilitates the movement of domain walls when polarized [11]. Simultaneous doping of A-site and B-site has been also carried out on other perovskite-structured compounds, resulting in changes in microstructure and properties. Some studies results show that ferroelectric transition temper-ature and electrical properties of oxide compounds are improved by controlling the concentration of defects and reducing leakage current during polarization [12,13].

Therefore, in this work, the BCZT-*x*(Nb + Y), (*x* = 0 mol%, 0.05 mol%, 0.1 mol%, 0.2 mol%, 0.3 mol%) piezoceramics were prepared through traditional solid-state sintering method. Meanwhile, the effects of the Yttrium and Niobium oxides on piezoelectric properties were discussed. This study elucidates the influence mechanism of Y^3+^ and Nb^5+^ co-doping of A-site and B-site on phase structure, microstructure, and electrical properties of BCZT ceramics.

## 2. Material and Methods

### 2.1. Preparation

The (Ba_0.85_Ca_0.15_) (Ti_0.90_Zr_0.10_)O_3_ + *x* Y^3+^ + *x* Nb^5+^ (abbreviated as BCZT-*x*(Nb + Y), *x* = 0 mol%, 0.05 mol%, 0.1 mol%, 0.2 mol%, 0.3 mol%) lead-free piezoceramics were synthesized by conventional solid-state method. The barium carbonate (BaCO_3_, 99.8%, Chengdu Kelong Chemical Reagent Factory, Chengdu, China), calcium carbonate (CaCO_3_, 99.5%, Sinopharm Group Chemical Reagent Co., Shanghai, China), zirconium dioxide (ZrO_2_, 99.8%, Chengdu Kelong Chemical Reagent Factory), titanium dioxide (TiO_2_, 99.0%, Tianjin Comio Chemical Reagent Co., Tianjin, China), yttrium oxide (Y_2_O_3_, 99.9%, Shanghai Aladdin Reagent Company, Shanghai, China) and niobium pentoxide (Nb_2_O_5_, 99.9%, Shanghai Aladdin Reagent Company, China) were used the electronic balance with the accuracy of 1/10,000 for weighing, with the error not exceeding 0.0001 g. The weighed powder were mixed in ZrO_2_ ball-mill tanks containing different proportions for 24 h and dried. Then, the dried powders were pre-sintered at 1200 °C for 2 h to form the precursor powder. The pre-sintered powder was ball-milled again for 24 h, mixed with 5% paraffin, and then pressed into solid discs with 12 mm diameter and of 1.1–1.2 mm thickness. Before sintering, the adhesive was volatilized completely at 600 °C for 2 h, and then all samples were sintered in air at 1400 °C for 4 h, and then cooled with the furnace.

### 2.2. Characterization

XPS tester was used to test the elemental composition, valence state and specific substitution position of ceramic samples. The XRD tester was used to analyze the phase of BCZT based ceramic samples in this experiment. The equipment model is X ‘PERt-Pro, PANAlytical. ZEISS SUPRA40 scanning electron microscope (SEM) was operated to analyze the surface microscopic morphology of BCZT ceramic samples in this experiment, and the multiple of photography was 1 k–8 k times. It was mainly used to analyze the microstructure, grain distribution, grain growth size and defects of the ceramic samples. High-resolution images of lattice stripes and electron diffraction patterns were observed using a transmission electron microscope (JEM-1200EX) produced by JEOL, Japan.

In order to test electrical properties of BCZT-*x*(Nb + Y) ceramics, Ag paste was uniformly printed on both surface of the circular piece by silkscreen printing and sintered at 600 °C for 30 min. Then, the ceramics were polarized at a voltage of 3 kV/mm for 30 min at room temperature. The piezoelectric coefficient (*d*_33_) was tested with a quasi-static *d*_33_ tester (Model ZJ-3AN, Institute of Acoustics, Chinese Academy of Sciences). The electromechanical coupling coefficient (*k*_p_) was calculated from the 4294A impedance analyzer test (Agilent, Am). The dielectric constant (*ε*_r_) at room temperature is calculated by measuring the dielectric loss (tan*δ*) and capacitance (*C*_p_) with TH2618B capacitance tester. The dielectric temperature spectrometer (model: WK6500B) produced in the UK was used to measure the relative dielectric constant as a function of temperature at a specific frequency. The test temperature was set as −10 °C–200 °C and the frequency was 100 Hz, 1 kHz, 10 kHz and 100 kHz. The ferroelectric loop (P-E) of the ceramic samples was measured using the RT66A ferroelectric tester produced in the United States at 3 kV, 10 Hz and room temperature.

## 3. Result and Discussion

### 3.1. XPS (Defect Chemistry Analysis)

XPS measurements are performed in Figure 1a to investigate the effect of Y and Nb elements co-doping on the elemental composition, valence state and specific substitution position of ceramic samples. For testing, a 5 × 5 mm sample was cut and attached to the sample tray. Place the sample in the Thermo Scientific K-Alpha XPS instrument sample room. The sample is irradiated with a 200–4000 eV Ar ion beam prior to analyses, which is rastered over the surface to be analyzed. When the pressure of the sample chamber is less than 2.0 × 10^−7^ mbar, the sample is sent to the analysis chamber. The excitation source is Al Kα ray (hv = 1486.6 eV), the spot size is 400 μm, and the working voltage is 12 kV. The full-spectrum scanning energy is 150 eV, the step size is 1 eV, and the scanning range is 1 eV–1000 eV. The scanning energy of oxygen spectrum is 50 eV, the step size is 0.1 eV, and the scanning range is 525 eV–536 eV [14]. The test results show peaks Ba, Ca, Zr, Ti, Y, Nb, O, and C and demonstrate the presence of these elements [15]. Here, since the position of the C 1s peak is steered by the sample work function, the work function method is used to account for the binding energy of the vacuum degree of C1s, which is usually 289.58 eV in the oxide [14]. Since the carbon element of the wax removal process has been completely excluded, there is no influence of carbon absorption and residual gas pollution on the samples of each group in this experiment. For the samples of BCZT, the Ba 3d_5/2_ peak is observed around 778 eV and the Ba 4d is measured around 91.55 eV. The Ti 2p spectra consists 2p_1/2_ and 2p_3/2_ electron, respectively occurred around 451 eV and 461 eV. The conclusion confirms that BCZT particles form in the pure phase and the possibility of secondary phase generation is excluded [16].

However, for the samples of BCZT-*x*(Nb + Y), a peak of Y 3d appear near 153 eV, guessing that the Y ion has diffused into the Ba lattice. At the same time, the emergence of Nb 3d near 205 eV support the diffusion of Nb ions in positions of Ti. Then, the mechanism of action of Y and Nb on the phase composition and microstructure of BCZT is further demonstrated by oxygen elemental analysis.

As a typical point defect, oxygen vacancies have a non-negligible influence mechanism on the microstructure and properties of ceramics. In order to characterize the oxygen vacancy concentration of different Y and Nb doping BCZT samples, XPS spectra of O 1s peaks for the pure BCZT, BCZT-0.1 mol%Y, BCZT-0.1 mol%Nb and BCZT-0.1 mol% (Y + Nb) are measured (shown in Figure 1b–d). Two binding energy peaks were obtained by peak-splitting fitting of oxygen spectra. Where, the peak of 529 eV corresponds to lattice oxygen, and the peak of 531 eV corresponds to oxygen vacancy. The area under the peak represents the oxygen vacancy concentration [14,17,18]. From the analysis of the Figure 1b, the relative concentration of oxygen vacancies increases when 0.1 mol% Y^3+^ is doped into BCZT. When 0.1 mol% Nb^5+^ is doped into BCZT, the relative density of oxygen vacancies decreases slightly. This phenomenon fully demonstrates that Y-doping introduced oxygen vacancies in BCZT, while Nb suppress the oxygen vacancy concentration. In ABO_3_, the M^3+^/M^2+^ ions with smaller ionic radius may occupy two sites. According to the tolerance factor principle, t = (r_A_ + r_O_)$\sqrt{2}$ (r_B_ + r_O_) (r_A_, r_B_ and r_O_ are ionic radii of A site cations, B site cations and oxygen anion, respectively), the A site is substituted by large ions [r (R^n+^) > 0.094 nm], and the B site is substituted by small ions [r (R^n+^) < 0.0087 nm]. For BCZT-*x*(Nb + Y) ceramic system, r (Ba^2+^) = 1.35 Å, r (Ti^4+^) = 0.605 Å, r (Y^3+^) = 0.90 Å, r (Nb^5+^) = 0.69 Å. because r (Ti^4+^) < r (Y^3+^) < r (Ba^2+^). As an amphoteric dopant, Y can replace the B site first, and then replace the A site after the B site saturated [15,19]. Therefore, when the doping amount is low, the low-valence Y^3+^ occupy the B site first. The charge difference between Ti^4+^ and Y^3+^ ions reduce the Coulomb force of the B-site, resulting in the generation of oxygen vacancies [20], as shown in Equation (1). In the same way, the high-valence Nb^5+^ act as donor to occupy the B site and is electrically compensated by electrons, as shown in Equation (2), making the oxygen vacancy concentration decrease. When Y and Nb are co-doped, the overall oxygen vacancy concentration does not change much. That’s exactly what we have seen here in the Figure 2.
(1)$$Y_{2}O_{3}+2{\mathrm{BaTiO}}_{3}\Rightarrow 2{\mathrm{Ba}}_{\mathrm{Ba}}^{\times }+2Y_{\mathrm{Ti}}^{\prime }+5O_{O}+V_{O}^{••}+2{\mathrm{TiO}}_{2}$$
(2)$$2{\mathrm{Nb}}_{2}O_{5}+4{\mathrm{BaTiO}}_{3}\Rightarrow 4{\mathrm{Ba}}_{\mathrm{Ba}}^{\times }+4{\mathrm{Nb}}_{\mathrm{Ti}}^{•}+12O_{O}+O_{2}\left(g\right)+4{\mathrm{TiO}}_{2}+{4e}^{\prime }$$

For higher concentrations of Y (*x* > 0.1 mol%), the Y^3+^ replace only the A site and is charged by an electron, resulting in a reduction in oxygen vacancy, as shown in Equation (3).
(3)$$2Y_{2}O_{3}+4{\mathrm{BaTiO}}_{3}\Rightarrow 4{\mathrm{Ti}}_{\mathrm{Ti}}^{\times }+4Y_{\mathrm{Ba}}^{•}+12O_{O}+4\mathrm{BaO}+O_{2}\left(g\right)+{4e}^{\prime }$$

From the above observations, it can be concluded that doping ions can change the concentration of oxygen vacancy in BCZT ceramics. Appropriate oxygen vacancy is conducive to mass transfer effect to a certain extent [21]. The surface defects such as oxygen vacancies increase the active sites of activation energy, promote the interdiffusion of grain boundaries, and facilitate the movement of small ferroelectric domains and the formation of phase coexistence structures [22]. Meanwhile, grain boundaries and pores are squeezed, contributing to the formation of dense microstructure, which is conducive to the improvement of electric properties. However, too low oxygen vacancy concentration is not conducive to the improvement of electrical performance and too high will lead to deterioration of performance.

### 3.2. Phase and Structure

Figure 2(a1) shows the X-ray diffraction (XRD) patterns of BCZT-*x*(Nb + Y) ceramics. Three different standard phases of BCZT are shown with vertical lines in the Figure 1, which are PDF No.85-0386, PDF No.81-2200 and PDF No.0626 respectively, representing phase orthorhombic (O, Amm2) tetragonal (T, P4mm,) and rhombohedral (R, R3m) in turn. It can be observed in Figure 2(a1) that all ceramic samples have uniform peaks and typical perovskite structure, indicating that Ba^2+^, Ca^2+^, Ti^4+^, Zr^4+^, Y^3+^ and Nb^5+^ are incorporated into the oxygen octahedral lattice, forming a stable perovskite structure. Secondly, the partially magnification of XRD patterns are shown as Figure 2(a2), preliminarily determined that the sample corresponds to O, R, T three-phase. Notably, Figure 2(a1) shows diffraction peaks near 30°, 53°, and 63° that are not part of BCZT and are enlarged as shown in Figure 2(a3). Using Jade Conduct phase recovery analysis, these peaks were matched the PDF card (PDF # 24-1042) and verified as Barium Yttrium Niobium Oxide (Ba_2_YNbO_6_). Other peaks of Ba_2_YNbO_6_ were also compared with the standard PDF card (PDF # 24-1042), and the results were consistent. The lattice parameter calculation model of the double perovskite structure Ba_2_YNbO_6_ is as shown in Figure 2(b1,b2). Ba_2_YNbO_6_ is a typical cubic system crystal double perovskite oxide, Y^3+^ and Nb^5+^ are arranged in the center of oxygen octahedron, respectively [23,24]. It is reported that double perovskite oxide has ferromagnetic, ferroelectric and other physical properties [24].

To further illustrate the phase composition of BCZT-*x*(Nb + Y) ceramics, The XRD patterns of all samples were refined by synchrotron XRD Rietveld and the refinement parameters, phase fractions, and error values were obtained (listed in Table 1). Figure 2c shows the synchrotron XRD Rietveld refinement results of the *x* = 0.1 mol% samples. This result confirmed the coexistence of R-O-T three phases, and the phase fractions of O, R, and T were further obtained. By comparing the XRD Rietveld refinement results of all samples, it was further confirmed that polyphase coexisted in each group of samples, and the phase fractions of each group of samples was further shown in Figure 2d. In addition, as shown in Figure 2e, the lattice constant c/a increases abnormally at *x* = 0.1 mol%, reflecting the highly asymmetric structure and large lattice distortion of the oxygen octahedral. Considering that *d*_33_ is significantly increased when *x* is 0.1 mol%, it is not difficult to draw the conclusion that the increase of c/a is conducive to improve the piezoelectric properties. After Y^3+^ and Nb^5+^ enter the lattice, it is believed that their influence on c/a mainly reflected in the following two aspects: Firstly, when Y^3+^ and Nb^5+^ enter the perovskite lattice through B-site doping, the spatial structure effect caused by the radius difference of Y^3+^ and Nb^5+^ reduces the lattice symmetry, which is manifested as lattice distortion of oxygen octahedron. Secondly, a double perovskite structure is formed in BCZT, which increases c/a. Table 2 shows the crystal structure parameters of Ba_2_YNbO_6_ refined by the XRD refinement results. The results indicate that the crystal plane spacing of Ba_2_YNbO_6_ unit cell in (100) plane spacing is 8.43 Å, and each double perovskite unit cell (as shown in Figure 2(b1)) contains two perovskite lattices. The lattice constant of each perovskite lattice is about 4.21 Å, which is greater than the lattice constant (about 4.03 Å) of BCZT, which is in accordance with Zhou et al. [24]. It can be proved that the double perovskite structure Ba_2_YNbO_6_ formed by Y^3+^ and Nb^5+^ increases c/a. In the Ba_2_YNbO_6_ double perovskite structure, YO_6_ and NbO_6_ will also have some tilt, which will reduce the symmetry of the double perovskite structure. Moreover, the inhomogeneity of the structure between YO_6_ and NbO_6_ is an important reason for the formation of local stochastic strain, which further promotes polarization rotation and enhances piezoelectric effects [5].

### 3.3. Microscopic Structure

As we know, the properties of piezoelectric ceramics can also be reflected by micromorphology analysis. The surface microscopic topography of BCZT-*x*(Nb + Y) ceramics is interpreted in Figure 3(a1–a5). It can be clearly deduced that the average grain size decreases with the increase of the doping amount *x*. This phenomenon can be explained as Nb^5+^ and Y^3+^ inhibited the growth of BCZT grains. With the increase of *x*, grain boundaries are squeezed and grains become more and more compact (see Figure 3(a1–a5),e). It is worth noting that when *x* = 0.3 mol%, the grain development is abnormal and new grains are produced. This may be due to the ABO_3_ structure’s solution limit. When the doping amount exceed the solution limit of ABO_3_ [25], grains will be precipitated at grain boundaries and the growth of grains will be inhibited. Another way of saying it is that titanium is easily enriched at the grain boundary to form impurity phases (such as BaTi_2_O_5_ and BaTi_3_O_7_) during the sintering process [16]. During the cooling process, the impurity phase is converted back to BaTiO_3_, and then plenty of oxygen vacancies and barium vacancies are formed. Oxygen vacancies tend to accumulate at grain boundaries rather than inside grains. Good thermal stability of the grain boundary oxygen vacancy is thought to hinder grain growth [25].

TEM measurements were performed on *x* = 0.1 mol% ceramic samples to explore its crystal structures (see Figure 3(c1–c5)). Figure 3(c1) shows the measured region near the grain boundaries, and we have selected the region shown in A for magnification. The selected region electron diffraction (SAED) is shown in Figure 3(c2). The inerratic electron diffracted atomic arrangement in Figure 3(c2) reflects the good crystallinity of BCZT-0.1 mol% (Nb + Y) ceramics. Moreover, the high-resolution crystal plane arrangement is shown in Figure 3(c3), which can be seen that the crystal lattice spacings of 0.404 nm and 0.283 nm correspond to (011)_O_ and (110)_R_ crystal planes, respectively [26], which certifies the coexistence of the O and R phases. Figure 3(c4,c5) is a fast Fourier transform (FFT) image showing a regular arrangement of electron diffraction points, proving that the ceramic is a typical single crystal.

The macroscopic density of ceramics is closely correlated with the microstructure, grain size and other factors. The variations of the density and average grain size of BCZT-*x* mol% (Y + Nb) ceramics are shown in Figure 3d,e. As can be seen from the Figure 3d that when the content of Y and Nb is 0.3 mol%, the density of ceramic is the highest, up to 5.58 g/cm^3^. This may be caused by grain refinement and proliferation, resulting in a decrease in porosity and an increase in density.

The microscopic SEM, TEM and macroscopic density test analysis show that the samples with appropriate Y^3+^ and Nb^5+^ co-doped have promoted the formation of dense microstructure, which is conducive to grain boundary migration and providing an important basis for improving the macroscopic electrical properties.

### 3.4. Electrical Properties

The P-E hysteresis loops measured for BCZT-*x*(Nb + Y) ceramics at 3 kV, 10 Hz and room temperature as a function of the electric field are shown in Figure 4(a1–a5). *P*_r_ reaches its maximum at *x* = 0.1 mol% and then decreases with the addition of Y and Nb ion content as shown in Figure 4b.

This phenomenon can be interpreted by the following viewpoints. Firstly, there is a quasi-homotype phase boundary in which R, O and T phases coexist in the sample, and spontaneous polarization is easy to convert along all possible directions provided by the R-O-T three phases, resulting in an increase in *P*_r_ [8]. Secondly, the double perovskite structure Ba_2_YNbO_6_ is formed in the sample, and the lattice constant increases and the lattice symmetry decreases, resulting in highly asymmetric structure and large lattice distortion inside the crystal. Moreover, the inhomogeneity of the structure between YO_6_ and NbO_6_ is also crucial reasons for the local random strain, which further promotes the polarization rotation. Thirdly, the doping of trace Y and Nb makes the sample obtain more dense and full grains. The formation of ceramic dense structure and relatively full grain size may increase the degree of freedom of ferroelectric domain motion and promote the rotation of 180^o^ ferroelectric domain wall, which is conducive to obtaining high dipole polarization and reducing leakage current, which is favorable to the improvement of ceramic properties [27]. These are the reasons for the increase in ferroelectric properties at the beginning. Additionally, it can also be shown that the *P*_r_ reduces with the increasing of doped elements. The first reason is that the substitution of Y^3+^ at the B site reaches saturation and the excess Y^3+^ begins to replace the A site, reducing the lattice constant. And the substitution of Zr^4+^/Ti^4+^ by Nb^5 +^ ions cause oxygen vacancies at the B position, forming a defective dipole, which hinders the movement of the ferroelectric domain wall [27] Another reason is that the domain wall increases caused by grain refinement [28]. Therefore, 0.1 mol% Y and Nb ions are suitable for doping.

The change of piezoelectric and dielectric constant of BCZT-*x*(Nb + Y) ceramics are shown in Figure 4c,d. It can be found that the piezoelectric constant (*d*_33_), dielectric constant (*ε*_r_), and electromechanical coupling coefficient (*k*_p_) are all increased compared with the undoped BCZT ceramic, and reach the maximum value at *x* = 0.1 mol%. The dielectric loss (tan*δ*) generally decreases with the addition of Nb and Y ions. By doping Y and Nb ions, all BCZT-*x*(Nb + Y) ceramics are shown enhanced *d*_33_. The highest *d*_33_ is in the BCZT-0.1 mol% (Nb + Y) ceramic with the value of 667 pC/N, which is beyond the reach of ordinary elemental doping.

The following explanations are the main reasons affecting the trend of piezoelectric changes: (1) The introduction of a new Ba_2_YNbO_6_ phase with double perovskite structure improves the microstructure. The piezoelectricity of ceramics is closely related to micromorphology and density [29]. According to the results of SEM, the grain size is relatively large and full, and there are few grain boundaries in the sample, which reduces the obstacle of domain wall movement, thus improving the piezoelectric property [30]. (2) Phase structure. According to previous analyses, the R-O-T phases is coexisted in *x* = 0.1 mol% and *x* = 0.2 mol% samples. Due to the small number of crystal orientations in the single-phase structure, it has fewer polarization directions than the multiphase structure [31], resulting in more difficult domain conversion and movement in the single-phase structure than in the multiphase structure [32].

Furthermore, by comparing the research progress of BCZT-based piezoelectric ceramics in recent years [4,6,7,8,9,10], it can be found that the BCZT-*x*(Nb + Y) ceramics have excellent piezoelectric properties and *T*_C_ (see Figure 4d). In conclusion, when *x* = 0.1 mol%, the ceramic sample obtains the best comprehensive electrical properties, where *d*_33_ = 667 pC/N, *k*_p_ = 0.58, *ε*_r_ = 5656, tan*δ* = 0.022, *P*_r_ = 12.8 μC/cm^2^, *E*_C_ = 2.17 kV/cm.

Figure 5a−e shows the permittivity of BCZT-*x*(Nb + Y) frequency as a function of temperature at 100 Hz, 1 kHz, 10 kHz, and 100 kHz. It is known to all that piezoelectric materials have temperature dependence as the temperature increases. Different phase transitions will be experienced, including tripartite phase (R phase), orthogonal phase (O phase), tetragonal phase (T phase), cis-electric cubic phase (C phase), etc. The abnormal peak of *ε*_r_-*T* curve in the figure corresponds to R-O, O-T and T-C phase transitions from left to right. Curie temperature *T*_C_ is the critical temperature for the structural transition from T phase to C phase of piezoelectric materials, which also refers to the temperature from ferroelectric phase to cis-electric cubic phase. It is clear from Figure 5 that the O-T phase co-exists in each set of samples [5,24]. In addition, as shown in Figure 5c,d, an abnormal peak of R-O phase transition is found when *x* = 0.1 mol% and *x* = 0.2 mol%, indicating the coexistence of R-O-T three-phase structure at 0–40 °C in these two groups of samples, which also explains the excellent electrical properties of these two groups of samples. It can be seen that the distribution of the three phases is consistent with the results of XRD Rietveld refinement. Phase transition temperatures of all samples is shown in Figure 5f, which shows the coexistence region of three phases at room temperature.

In the ferroelectric phase region, the piezoelectric component is spontaneously polarized and has piezoelectric properties, while it does not have piezoelectric properties in the cis-electric phase region. In order to ensure that the piezoelectric components do not lose their piezoelectric properties due to heating when used, it is usually necessary to increase the Curie temperature of the piezoelectric material. As shown in Figure 5c, the *T*_C_ of the BCZT−0.1 mol% (Nb + Y) sample rises to 92.1 °C. This is due to the fact that dopants can effectively promote the growth of grains and contribute to the release of stress within the grains, thereby increasing *T*_C_ [33]. However, excess Y^3+^ and Nb^5+^ doping may reduce the stability of the oxygen octahedron [34], leading to a decrease in the Curie temperature. What is noteworthy is that Figure 5 also shows the shifting of the *T*c peak with increasing frequency. It is not difficult to observe from Figure 5 that with the increase of frequency, the corresponding *ε*_r_ value of samples in each group near the Curie temperature decreases, and the width of the Curie temperature peak becomes wider. This phenomenon reflects the diffuse phase transition behavior the dielectric constant, which is one of the typical characteristics of the relaxed ferroelectrics [35]. Based on this discovery, the dielectric relaxation properties of BCZT samples are further studied.

### 3.5. Dielectric Relaxation Behavior

Figure 6 shows the relationship of 1/*ε*_r_ and temperature for the BCZT-*x*(Nb + Y) samples at 100 kHz. Some parameters relevant to dielectric relaxation properties are given in Table 3, in which *T*_cw_, *T*_B_ and *T*_m_ respectively represent the Curie-Weiss temperature, the starting temperature conforming to the Curie-Weiss law and the temperature when the permittivity reaches its maximum value. Above *T*_cw_, the 1/*ε*_r_ value of the general ferroelectric body should have a linear relationship with the temperature, that is in accordance with Curie’s temperature law (as shown in functional Equation (4)) [35]:(4)$$\frac{1}{\varepsilon }=\left(\frac{T-T_{cw}}{C}\right)\;\left(T>T_{m}\right)$$

However, as shown in Figure 6, 1/*ε*_r_ is not linear with temperature in the range of *∆T*_m_, where *∆T*_m_ is the temperature range where the curve deviates from the Curieweis law. Therefore, such ferroelectrics that do not conform to Curieweis’s law are called relaxant ferroelectrics. The 1/*ε*_r_ and temperature of the relaxation ferroelectric conforms to the optimized Curie-Weiss law, as in Equation (5) [35]:(5)$$\frac{1}{\varepsilon }-\frac{1}{\varepsilon_{m}}=\left(\frac{T-T_{m}}{C}\right){\;}^{\gamma }\left(T>T_{m},\;1<\gamma <2\right)$$

The range of *γ* when the ceramic sample is a relaxant ferroelectric body is 1 < *γ* < 2. The fitting line in the insets of Figure 6 is obtained through the processing of Equation (5), which reveal the effect of the relationship between ln(1/*ε* − 1/*ε_m_*) and ln(*T* − *T_m_*) at 100 kHz of samples, and the value of diffusion coefficient γ can be defined by the slope. A larger value of *γ* in the range of 1–2 indicates a wider phase transition temperature in each microregion and therefore a stronger diffuse phase transition. The inset of Table 3 and Figure 6 shows that the strongest dielectric relaxation behavior is observed at the sample with *x* = 0.05 mol%. Due to the co-doping of Y^3+^ and Nb^5+^ in different valence states, lattice distortion and charge imbalance will occur in the perovskite lattice [36], and polar nano regions will form in the ceramic sample, thus promoting dielectric relaxation behavior. Zhao et al. researched that piezoelectric ceramics with large relaxation behavior can improve their dielectric and energy storage properties [37]. The above analysis results indicate that BCZT-*x*(Nb + Y) ceramic samples with strong relaxation properties have better electrical properties.

## 4. Conclusions

In this work, combined with defect analysis, phase coexistence mechanism, double perovskite structure model and microstructure analysis, the BCZT-*x*(Nb + Y) lead-free piezoelectric ceramics were designed and fabricated by the traditional solid-phase reaction method. A BCZT-*x*(Nb + Y) ceramic sample with ultra-high voltage electrical properties (*d*_33_ = 667 pC/N, *k*_p_ = 58%) was obtained at *x* = 0.1 mol%. XPS defect chemistry analysis, XRD phase analysis and TEM results together showed the Ba_2_YNbO_6_ phase with double perovskite structure is formed in the ceramics, resulting in reduced crystal symmetry and the lattice distortion of the ceramics is more significant because of its larger lattice constant. In addition, XRD, TEM and Graph of dielectric-constant as a function of temperature showed that R-O-T phases are found to coexist when *x* = 0.1 mol% and *x* = 0.2 mol%, forming a multiphase coexistence structure with more spontaneous polarization directions. SEM analysis showed that the ceramic formed dense and full grain. The above analyses together indicate the reason why the ceramic samples have good piezoelectric properties. This work provides not only a leadfree piezoelectric material with high performance, but also an idea to construct a double perovskite structure within Pb-free piezoelectric ceramics for improving the piezoelectric properties, which could be instructive for the future design of piezoelectric ceramics with higher performance. Nevertheless, only using the traditional solid phase method has the limitation of low phase formation rate. Thus, the hydrothermal method and sol-gel method can be used to prepare the precursor of double perovskite structure to improve the formation rate of double perovskite structure. In the future, other kinds of double perovskite structures can be introduced to further explore their influence mechanism on BCZT.

## Figures and Tables

**Figure 1 materials-16-04044-f001:**
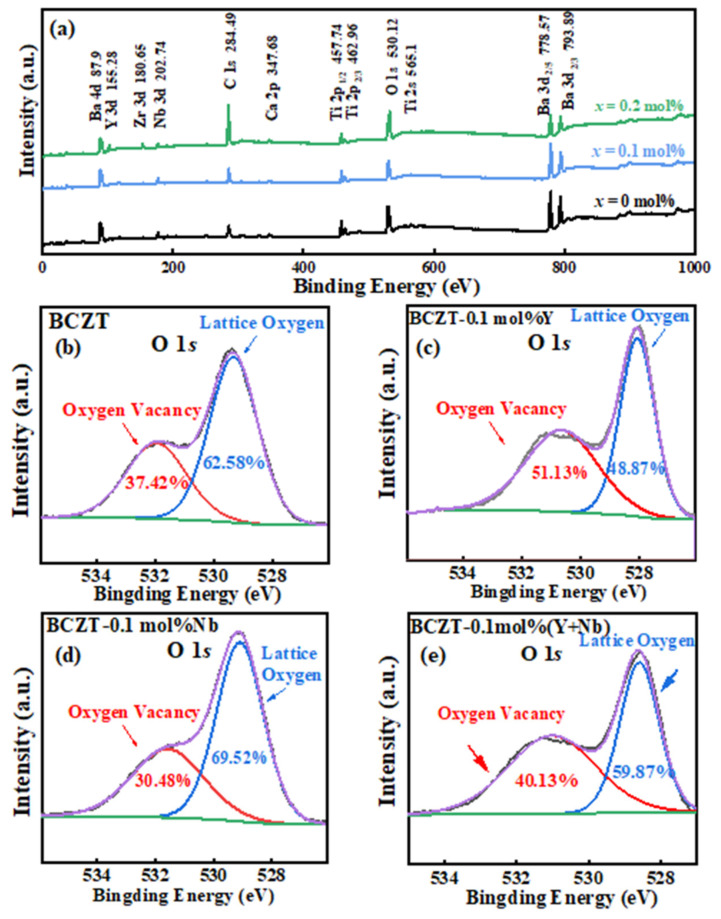
(**a**) the XPS results of BCZT and BCZT-*x*(Nb + Y) samples. (**b**–**e**) XPS spectra of O 1s peaks binding states for BCZT-*x*(Nb + Y).

**Figure 2 materials-16-04044-f002:**
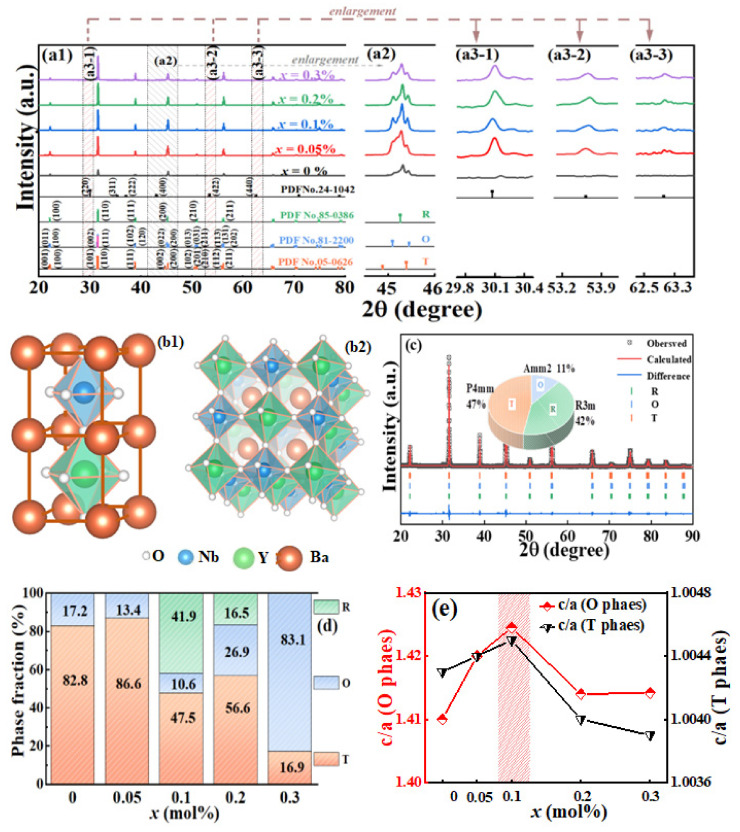
(**a1**–**a3**) XRD patterns of BCZT-*x*(Nb + Y) ceramics. (**b1**,**b2**) Ba_2_YNbO_6_ Schematic diagram of double perovskite structure (**c**) XRD Synchrotron Rietveld refinement of the *x* = 0.1 mol% ceramics. (**d**) phase fractions of BCZT-*x*(Nb + Y) ceramics. (**e**) c/a of BCZT-0.1 mol%(Nb + Y) ceramics from the Rietveld refinement.

**Figure 3 materials-16-04044-f003:**
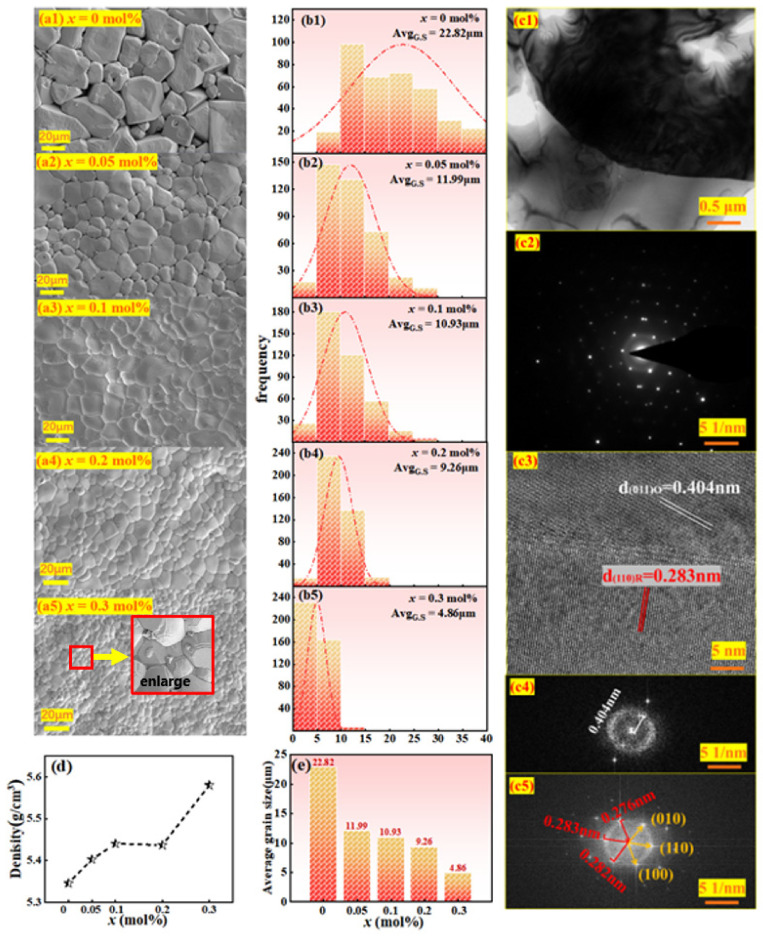
(**a1**–**a5**) SEM microscopic topography of BCZT-*x*(Nb + Y) ceramics, (**b1**–**b5**) grain size distribution of ceramics, (**c1**–**c5**) TEM of x = 0.1 mol% ceramics, (**c1**) topography, (**c2**) selected region electron diffraction, (**c3**) high-resolution lattice fringes. (**c4**,**c5**) fast Fourier transform of (**c3**). (**d**) density for BCZT and BCZT-*x*(Nb + Y). (**e**) average grain size distribution of BCZT-*x*(Nb + Y) ceramics.

**Figure 4 materials-16-04044-f004:**
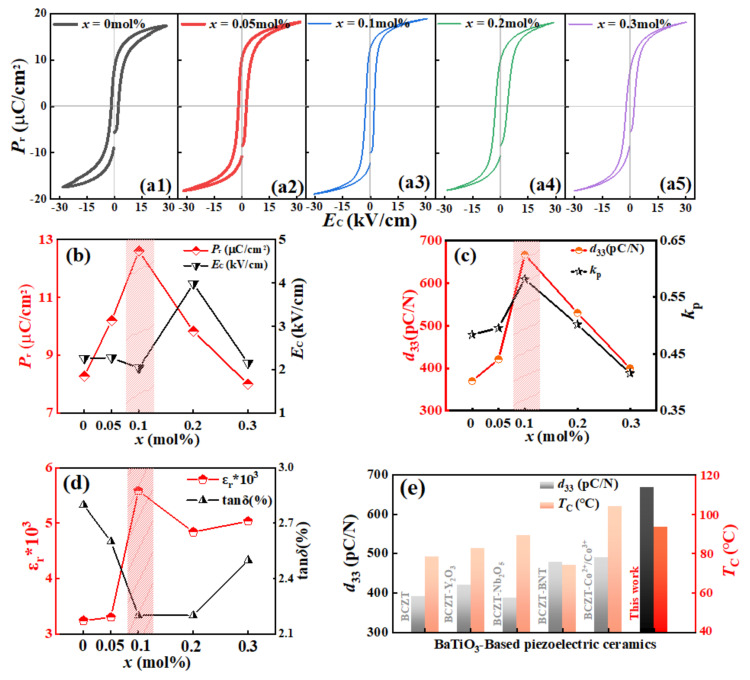
(**a1**−**a5**) P-E hysteresis loop of BCZT-*x*(Nb + Y) ceramic at 3 kV, 10 Hz and room temperature. (**b**) The *P*_r_ and *E*_C_. (**c**) *d*_33_ and *k*_p_, (**d**) *ε*_r_ and tan*δ* of BCZT-*x*(Nb + Y) ceramic samples. (**e**) Comparison of the *d*_33_ and *T*_C_ in reported BCZT-based piezoelectric ceramics in this work [4,6,7,8,9,10].

**Figure 5 materials-16-04044-f005:**
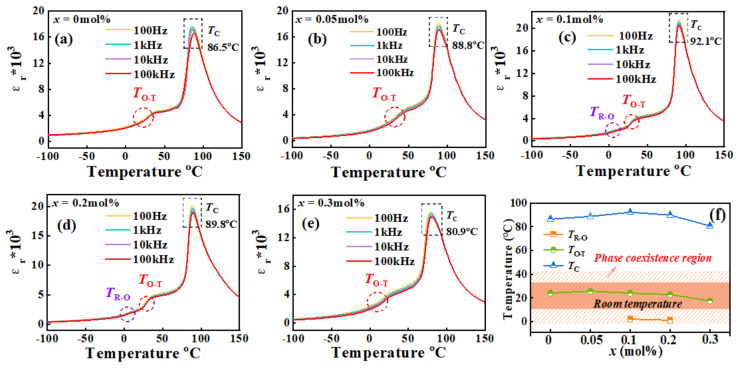
(**a**–**f**) The relationship between temperature and dielectric constant of BCZT-*x*(Nb + Y) ceramic samples.

**Figure 6 materials-16-04044-f006:**
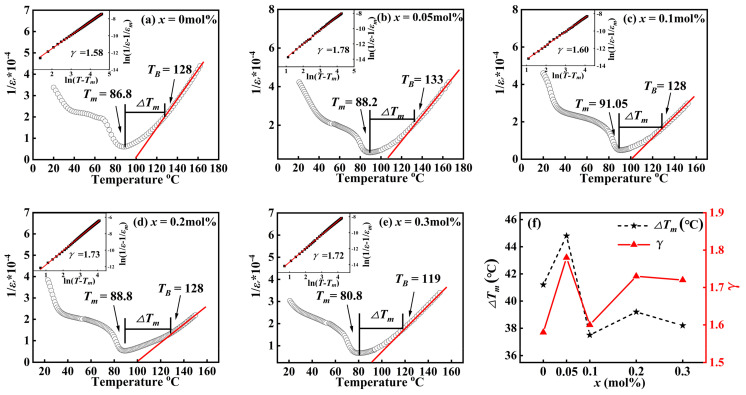
(**a–f**) The 1/*ε*_r_ and temperature for the BCZT-*x*(Nb + Y) samples at 100 kHz. Insets is ln(1/*ε* − 1/*ε_m_*)/ln(*T* − *T*_m_).

**Table 1 materials-16-04044-t001:** Crystal structure parameters of BCZT-*x*(Nb + Y) piezoceramics.

*x* (%)	Phase	a (Å)	b(Å)	c (Å)	c/a	α = β = γ (°)	V (Å^3^)	Fraction (%)	R_p_ (%)	R_wp_ (%)
0	O	4.0295	5.6824	5.6872	1.4113	90.00	130.223	17.1	7.66	9.59
	T	4.0000	4.0000	4.0170	1.0043	90.00	64.272	82.8		
0.05	O	4.0259	5.7085	5.7169	1.4200	90.00	131.38	13.4	8.37	10.88
	T	4.0007	4.0007	4.0183	1.0044	90.00	64.315	86.6		
0.1	R	4.0014	4.0014	4.0014	1.0000	89.91	64.069	41.9		
	O	4.0537	5.7570	5.7746	1.4245	90.00	134.761	10.6	8.41	11.20
	T	4.0012	4.0012	4.0192	1.0045	90.00	64.346	47.5		
	R	4.0096	4.0096	4.0096	1.0000	89.99	64.460	16.5	8.38	10.92
0.2	O	4.0127	5.6702	5.6741	1.4140	90.00	129.10	26.9		
	T	4.0015	4.0015	4.0176	1.0040	90.00	64.330	56.6		
0.3	O	4.0060	5.6637	5.6669	1.4142	90.00	128.57	83.1	7.52	10.53
	T	4.0017	4.0017	4.0175	1.0039	90.00	64.335	16.9		

**Table 2 materials-16-04044-t002:** Crystal structure parameters of Ba_2_YNbO_6_.

a (Å)	b (Å)	c (Å)	α (◦)	β (◦)	γ (◦)
8.43159	8.43159	8.43159	90	90	90

**Table 3 materials-16-04044-t003:** Some parameters extracting from Figure 6.

*x* (mol%)	*T_cw_* (°C)	*T_B_* (°C)	*T_m_* (°C)	*∆T_m_* (°C)	*ε_m_* (100 Hz)	*γ*
0	100.1	128	86.8	41.2	16,301	1.58
0.05	106.5	133	88.2	44.8	17,062	1.78
0.1	101.2	128	91.05	36.95	20,545	1.60
0.2	99.8	128	88.8	39.2	18,905	1.73
0.3	93.5	119	80.8	38.2	14,920	1.72

## Data Availability

Data is available from the corresponding author upon reasonable request.

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
