# Peer review of "Yttrium and Niobium Elements Co-Doping and the Formation of Double Perovskite Structure Ba2YNbO6 in BCZT"

_materials, 2023, doi:10.3390/ma16114044_

Round 1

Reviewer 1 Report

Dear Authors, 

I have some comments on your study:

- Figures shouldn't be placed at the beginning of the subsections, but after the citations;

- l. 56, ...., The BCZT - should be "the BCZT .., the same "However, For..." and others in the whole text;

- coating with a paste - do the coating process and its accuracy may have any influence on the process?

- the further directions of the research should be indicated in the conclusions,

- the strengths and limitations of the obtained results and applied methods should be described,

- were the experiments repeated to confirm the findings and for statistical reasons?

- x = 0 mol%, 0.05 mol%, 0.1 mol%, 0.2 mol% - were used in the experiments. Whether smaller differences in values could lead to better results?

Best, 

R

The English language is correct. Some corrections related to the writing should be made, e.g. l. 56, ...., The BCZT - should be "the BCZT .., the same "However, For..." and others in the whole text;

Author Response

Dear Reviewer

Thank you very much for taking your time to review our manuscript. We also want to express our appreciation for your valuable suggestions. Our response to your comments can be found in the attachment below.

Reviewer 2 Report

The paper entitled "Yttrium and Niobium elements co-doping and the formation of double perovskite structure Ba2YNbO6 in BCZT" by Runyu Mao and co. presents the enhancement of the  piezoelectric properties of BCZT lead-free piezoelectric ceramics by Y3+ and Nb5+ co-doping.

The paper is well written and the subject is interesting, but some minor revisions must be done, as follows:

1) Table 2 must be explained in the text

2) row 229 : "Fig. (c1) shows the measured region near the grain". It is Fig.3(c1)

The Quality of English Language is ok

Author Response

(The authors gave the same response as above.)

Reviewer 3 Report

This abstract describes an investigation into the electrical properties, dielectric relaxation behavior and microstructure of lead-free piezoelectric ceramics of the type (Ba0,85Ca0,15) (Ti0,90Zr0,10)O3 + x Y3+ + x Nb5+ (abbreviated as BCZT -x (Nb+Y), with x = 0%, 0.05%, 0.1%, 0.2%, 0.3%).

1.      There are no considerations for the effects of the used paraffin, as for likely uptake and influence of residual gas contamination. What is, for example, the O and H content of the processed samples? Did any hydroxide phase segregate to grain or phase boundaries?

2.      For the section material and methods, mentioned the binding energy calibration. As an alternative, you can try to use the work function method . The best is to obtain the work function of your sample by UPS in the same instrument. If you do not have that possibility use the literature value. This will likely result in smaller error than just using the conventional C 1s referencing.

3.      XPS experimental details are not complete. See the example of XPS protocol in section 6.1 of https://doi.org/10.1016/j.pmatsci.2019.100591. What was the electron emission angle, the size of analyzed area? Where samples sputter-etched prior to analyses? If so, what was the Ar+ energy and incidence angle? What was the base pressure during analyses? Was charge neutralizer used? All of these aspects are crucial for the correct interpretation of experimental results.

4.      I suggest to show elemental concentrations in a separate figure to make more room for the XPS spectra which are now hardly readable.

5.      The authors say, “The peak at 529 eV corresponds to the lattice oxygen and the peak at 531 eV is assigned to oxygen vacancies.” Maybe two or three papers should be added to clarify the peak around 531 eV (for example, https://doi.org/10.1039/D0CP01010C).

7.      What is the mechanism of action of Y and Nb on the phase composition and structure of BCZT, as demonstrated by the oxygen elemental analysis? Discussion is required around to the boundary interdiffusion effects on dielectric ( https://doi.org/10.1016/j.apsusc.2019.07.003).

Author Response

(The authors gave the same response as above.)

Reviewer 4 Report

This manuscript presents the synthesis of double-perovskite Ba2YNbO6 modified with Y/Nb ions and discusses its detailed electrical properties. The sample is well characterized, and the data is explained well. However, some following issues should be addressed before accepting in Materials.

1.     The author should add a sentence in the introduction that simultaneous doping of A-site and B-site has been also carried out on other perovskite-structured compounds to improve the electrical properties of oxide compounds. These articles can be cited: https://doi.org/10.1016/j.ceramint.2022.01.307 ; https://doi.org/10.1016/j.jallcom.2020.156131

2.     For phase and structure discussion:

-       the author mentions the coexistence of three-phase structures of O, R, and T but did not explain the name of each phase.

-       The author shows “Table 1. Crystal structure parameters of BCZT-0.1 mol%(Nb+Y) piezoceramics”. The caption is specific for a sample with 0.1mol% doped, but the data show all compositions Check the possible error for this caption.

-       Fig 2c did not mention in the discussion. Add the sentence the profile of Rietveld refinement represented by BCZT-0.1 mol%(Nb+Y) sample.

3.     The author reports many parameters of dielectric data in Table 3 such as Tcw, TB, Tm, and ∆Tm. Again, author should explain the name of the parameters before giving the symbol. Author should also discuss more this data.

4.     In my opinion, Fig. 5 also shows the shifting of the Tc peak with increasing frequency. Author only explained the diffuse phase transition behavior supported by the increase in γ value. However, the shift in the Tc peak also clearly describes the relaxor properties, not only DFT. It would be interesting to discuss the appearance of the relaxor properties of these compounds and also add to the discussion the effect of increasing dopant composition induces the appearance of these properties more clearly.

5.     Some error corrections in the manuscript:

·      Writing the cell unit parameters or other symbols should use italic text, and rechecked all these symbols.

·      Page 1 line 42, only uses “lower” instead of “much lower” for Tc 74 oC compared with BCZT of 85oC

·      In the first paragraph of the introduction, the author should mention the nominal formula (Ba0.85Ca0.15)(Ti0.90Zr0.10)O3 then give the denotation of (BCZT).

·      Adding the trademark of the oxide precursor used in the preparation section.

·      Page 2 line 69, revise the writing of zro2. Rechecked all typo errors in the manuscript.

·      The author should be consistent to use “hours” or “h” in the whole manuscript (page 2 ln 73).

·      XPS instrument needs to write the brand model.

·      Writing of XPS peaks (i.e., 3d5/2, Y 3d, and others) on the discussion text needs to use subscripts. Recheck and revise.

Author Response

(The authors gave the same response as above.)

Round 2

Reviewer 4 Report

I am satisfied that the authors revise the manuscript as per the suggestions of the reviewers, now it can be accepted for publication in Materials.